Thermal ecology of the Mexican Garter Snake (Thamnophis eques): temporal and spatial variations

http://orcid.org/0000-0002-4290-0838 Venegas-Barrera Crystian S. 1
http://orcid.org/0000-0003-4685-5322 Sunny Armando 2
http://orcid.org/0000-0003-3053-3257 Manjarrez Javier 3 jsilva@uaemex.mx
1 Instituto Tecnológico de Ciudad Victoria, Tecnológico Nacional de México , Ciudad Victoria, Tamaulipas , Mexico
2 Centro de Investigación en Ciencias Biológicas Aplicadas, Universidad Autónoma del Estado de Mexico , Toluca, Estado de México , Mexico
3 Facultad de Ciencias, Universidad Autónoma del Estado de México , Toluca, Estado de México , Mexico
Hedrick Ann
Electronic publication date: 2025 Jan 13
Publication date: 2025
Volume: 13
Electronic Location ID: e18641
Received 2024 Aug 15; Accepted 2024 Nov 13
Copyright: © 2025 Venegas-Barrera et al.
Copyright year: 2025
Copyright holder: Venegas-Barrera et al.
License: This is an open access article distributed under the terms of the Creative Commons Attribution License, which permits unrestricted use, distribution, reproduction and adaptation in any medium and for any purpose provided that it is properly attributed. For attribution, the original author(s), title, publication source (PeerJ) and either DOI or URL of the article must be cited.
License URL: https://creativecommons.org/licenses/by/4.0/

Keywords: Body temperature, Forest, Grassland, Repeated measures ANOVA, Thamnophis eques

Funding: Universidad Autónoma del Estado de Mexico 1425/2000 CONAHCYT This work received financial support from the Secretary of Research and Advanced Studies (SYEA) of the Universidad Autónoma del Estado de Mexico, through a grant No. 1425/2000. Crystian S. Venegas-Barrera received a scholarship from CONAHCYT. The funders had no role in study design, data collection and analysis, decision to publish, or preparation of the manuscript.

==============================
Heterogeneous environments provide different daily and seasonal thermal conditions for snakes, resulting in temporal and spatial variations in body temperature (Tb). This study analyzes the Tb of Thamnophis eques in the forest and grassland of a Mexican locality through daily and seasonal profiling. The patterns were obtained from seminatural enclosures in the field with a point sampling strategy to analyze temporal and spatial variations in Tb. The variation of Tb throughout the day was correlated with air and substrate temperatures, both in the grassland and in the forest. The average Tb in the grassland was 0.88 °C greater than in the forest. Our results indicate that T. eques showed differences in Tb between grassland and forest, principally in late spring and early summer during the early rise and late plateau phases, coinciding with the presence of foliage on the willow trees present in the study area, while in late summer and early autumn, the Tb was similar when willow trees canopy cover was absent (May–September). Our results support the hypothesis that the Tb of snakes differ between forests when the willows have leaves and is similar when canopy cover is equivalent, in this case, when tree canopy cover was absent. Our results also shown that T. eques presented daily and seasonal warming patterns similar to other Arizona populations and like those of other northern Gartersnakes. However, this result may not be valid for the entire wide distribution and consequent diversity of habitats of T. eques. The information of Tb in T. eques through daily and seasonal profiles in different habitats could aid in understanding the effects of environmental conditions on the ecological strategies deployed by snakes on habitat selection.

Introduction

The thermal environment is one of the most important constraints that affects ectotherms habitat selection (George, Connette & Thompson FR, 2017; Nadeau, Urban & Bridle, 2017; Johnson, Poulin & Somers, 2022). For instance, snake performance is sensitive to body temperature (Tb) in relation to temporal and spatial environmental thermal fluctuations (Shine & Lambeck, 1990; Taylor, DeNardo & Malawy, 2004; Leliévre et al., 2011; Figueroa-Huitrón et al., 2024a). Such thermal variations induce snakes to move and select resources between unfavorable and favorable thermal patches that influence their metabolic activity (Reinert & Zappalorti, 1988; Row & Blouin-Demers, 2006; Giacometti et al., 2021). These favorable thermal patches or microenvironments are chosen by individuals to initiate multiple activities, such as foraging, escaping predators, among others (Blouin-Demers & Weatherhead, 2001a; Rowe et al., 2022; Veselka et al., 2023), and offer different daily and seasonal thermal conditions, which produce temporal variations in Tb and affect snake behavior or activity (Reinert & Zappalorti, 1988; Blouin-Demers & Weatherhead, 2001a; Harvey & Weatherhead, 2010; Spaseni et al., 2024).

The thermal quality of the habitat reflects the limited windows of opportunity for daily and temporal activity of snakes (Hertz, Huey & Stevenson, 1993; Johnson, Poulin & Somers, 2022). For example, in Mexico, Thamnophis eques had two seasonal peaks of activity, from September to October and April to June (Manjarrez, 1998). Mexican Gartersnakes are active from 14.8–27.8 °C air temperature, 19.0–31.5 °C surface temperature, and 22.1–22.5 °C water temperature (Holm & Lowe, 1995). They are active diurnally and nocturnally and their activity on aquatic prey is correlated with the abundance of prey and the air and water temperatures (Drummond & Macias-Garcia, 1989).

The Tb of several snake species changes depending on the season and environment evaluated (Row & Blouin-Demers, 2006). For example, the Tb of the Mexican garter snake Conopsis biseralis increases from 15 °C at 9 am to 30 °C at 2 pm during the day (Raya-García & Arteaga-Tinoco, 2024), and a similar pattern occurs with the Australian Pseudechis porphyriacus, which, in the morning with a Tb of 16–18 °C, increases to 30 °C a few hours later. In spring, its Tb is on average 5 °C higher than that in summer (Shine & Lambeck, 1990). On the other hand, the Tb of the rattlesnake Sistrurus catenatus in June, with a Tb of 10 °C at 0 h, increased to 15 °C at 12 h, but two months later, in August, the increase was greater because it increased from 17 °C at 0 h to 25 °C at 12 h (Harvey & Weatherhead, 2010). However, few studies have explored changes in daily activity during different seasons in contrasting environments, which can improve our ability to predict the responses of species to natural or induced environmental changes (Rugiero et al., 2013; Jaramillo-Alba et al., 2020).

Thermal studies have shown that most snakes present a daily thermoregulation pattern characterized by three thermal body phases: (1) a rapid rise in the morning (08:00 to 12:00 h), (2) a stabilized phase at mid-day (plateau phase from 12:00 to 15:00 h), and (3) a drop phase in the afternoon (Seebacher & Franklin, 2005; Harvey & Weatherhead, 2010; Michel & Bonnet, 2010). The duration of each phase change depends on the season, environmental conditions, or latitude, from a pattern with equal duration phases to a pattern dominated by one or two phases (Shine, 1987; Peterson, Gibson & Dorcas, 1993). Species such as the rat snake (Elaphe obsoleta) show differential habitat selection depending on sex and the thermal properties of the environment (Carfagno & Weatherhead, 2006; Weatherhead et al., 2012). During the day, in eastern Ontario, black rat snakes are significantly warmer in barn habitats than in edge and forest habitats (Blouin-Demers & Weatherhead, 2002); however, forest edges offer the best opportunities for behavioral thermoregulation because they offer both refuge from high temperatures (forest) and warmest possible habitats (open habitats). Males showed the greatest affinity for edges, whereas females (both gravid and nongravid) used forests more because forests are the coolest of all available habitats for black rat snakes (Carfagno & Weatherhead, 2006; Blouin-Demers & Weatherhead, 2002). However, little is known about how temporal variations in the structure of vegetation at local scales can affect daily snake thermoregulation patterns.

Gartersnake species (Genus Thamnophis) are broadly used as a biological model for determining habitat use and thermal activity (Sprague & Bateman, 2018), but little is known about the daily and seasonal thermoregulatory patterns of southern gartersnake species (Rossman, Ford & Seigel, 1996; Langford, Borden & Nelson, 2011). One of these southern species is the Mexican Gartersnake (Thamnophis eques), a medium-sized generalist predator widely distributed across the Mexican Plateau, where it feeds on aquatic and terrestrial preys (Macías-García & Drummond, 1988; Manjarrez, 1998; Manjarrez, Pacheco-Tinoco & Venegas-Barrera, 2017). It has bimodal annual activity (Manjarrez, 1998), in April–June and September–October. In USA and México, the northern Mexican gartersnake is federally listed as threatened (SEMARNAT, 2010; Fish and Wildlife Service, 2023). This consideration is very important in Mexican T. eques populations that suffer a high and continuous rate of land use change (González-Fernández et al., 2018).

Thamnophis eques has a wide use of habitats, is found in river systems that flow through mesquite grasslands in deserts, forests, or gallery forests and through riparian vegetation around permanent water sources. This diversity of occupied habitats offers different environments and thermal opportunities to T. eques, making it a suitable biological model to analyze variations in their thermoregulation on contiguous environment, which may be mostly exposed to the sun, such as a grassland, or with less sun exposure, such as a forest (Rossman, Ford & Seigel, 1996; Manjarrez, 1998; Conant, 2003; González-Fernández et al., 2018; Sprague & Bateman, 2018). In these environments, Mexican Gartersnakes require vegetative cover because snakes are often located within 15 m from permanent water with lush vegetation (Greenwald, 2003; Sprague & Bateman, 2018), such as willow trees (Salix sp.), riparian deciduous trees that lose their leaves from late fall (November) to early winter (January; Saska & Kuzovkina, 2010).

Wetlands are important environments for Mexican Gartersnakes (Manjarrez, 1998; Conant, 2003; González-Fernández et al., 2018; Sprague & Bateman, 2018), because they offer a gradient of warmer and cooler conditions (Macías-García & Drummond, 1988), leading to a more attenuated and stable microenvironmental temperatures available for snakes (Brom & Pokorný, 2009). The seasonal loss of leaves from deciduous trees causes a reduction in the canopy, from late fall to early winter, increases the solar incidence on the ground and increases the environmental temperature, such as in open environments (Lara-Reséndiz et al., 2014; Weatherhead et al., 2012). Studies comparing the thermal environments offered to snakes by grasslands and forests have shown that open-canopy environments, such as grasslands, are thermally heterogeneous due to daily extreme fluctuations in surface temperature (Blouin-Demers & Weatherhead, 2001a, 2001b; Johnson, Poulin & Somers, 2022), while forests offer a fluctuating habitat with high spatial and thermal heterogeneity, giving snakes fast access to both warm and cool microclimates, which offers chances to modify body temperatures when air temperatures become intolerant (Blouin-Demers & Weatherhead, 2001b; Jaramillo-Alba et al., 2020).

The objective of this study was to quantify the environmental thermal quality of the population of T. eques by assessing the quality of two available environments, forest and grassland in Toluca Valley within the Central Mexican Plateau. At the same time, we studied the Tb in open-canopy grasslands in contrast to that in forest environments throughout daily and seasonal period of activity. We propose that in these environments, the Tb of snakes, such as T. eques, may differ between open environments (grassland) and forest environments characterized by perennial and deciduous trees that lose foliage in autumn. We tested two hypotheses: (1) the Tb of grassland is greater than that of forest during seasons when tree canopy cover is continuous (May–September), and (2) the Tb of grassland and forest are similar when tree canopy cover is absent (October–November). We employed a sampling-point strategy on mesh encloses in the field to analyze temporal and spatial variations in Tb, air and substrate temperature. This information could help to understand the effects of ecological strategies followed by snakes on habitat selection according to thermal environmental conditions (Weatherhead et al., 2012; Giacometti et al., 2021).

Materials and methods

Study area

The snake population is located at the El Cerrillo campus (99°41′W, 19°24′N Datum WGS84) of Universidad Autonoma del Estado de Mexico, 17 km NE of Toluca City, Mexico, at an altitude of 2,550 m. The area consists of a waterbody surrounded by a seasonally flooded grassland (W‒NW), characterized by Muhlenbergia sp. and Festuca sp., and a forest of Salix-Pinus (E‒SE). Environments offer different thermal conditions and are spatially separated by 200 m (Fig. 1). For example, during spring, the grassland offers a significantly higher thermal environment (mean air temperature 25 ± 1.97 °C) than the forest (20 ± 1.98 °C; Kolmogorov–Smirnov test Z = 3.32, P = 0.0008), and the same happens during winter (grassland 20.7 ± 1.66 °C; forest 16 ± 1.38 °C; Kolmogorov–Smirnov test Z = 3.32, P = 0.0001). In fall, willows tree canopy is absent (October–November), which increases the amount of solar radiation on the substrate (Lara-Reséndiz et al., 2014).

Figure 1 Localization of study area at the “El Cerrillo” campus of the Universidad Autonoma del Estado de Mexico, 17 km NE of Toluca City, Mexico.

Also shown are the experimental enclosures inside of the grassland and forest surrounding the waterbody, where we conducted warming trials of T. eques during the period of May to November. Background image taken from Google Earth Pro. Photo credit grassland and forest: Crystian S. Venegas-Barrera.

Field methods

We frequently captured snakes among grasses by searching, under trunks or rocks, near the shore of water bodies, and some were found simply basking on the substrate. All snakes were captured by hand. Snakes were individually housed in glass tanks (51 × 26 × 28 cm) containing water dishes and clay shelters, at an ambient room temperature of 20–25 °C and a natural light-dark photoperiod. They were fed twice a week with live fish until their release at the same capture site.

From March to November 1999, we characterized grassland and forest thermal environments, by recording air temperature (Ta), at 1.5 m above the ground, and substrate temperature (Ts), 2 cm above the ground; under shade every 5 min from 10:30 to 12:30 h (Ts ± 0.2; Ta ± 0.2 °C) at random points using a Schultheis quick registering thermometer.

All snakes were collected under a scientific collecting permit issued by the federal institution (FAUT-0188; Secretaría de Medio Ambiente y Recursos Naturales, México). This study received the approval of field permit (#07164; Secretaria del Medio Ambiente y Recursos Naturales, México) and the ethics committee of the Universidad Autónoma del Estado de México (Numbers 3589/2013SF, 4047/2016SF, and 4865/2019SF). All subjects were treated humanely in accordance with the basis of Mexican Federal Regulation for Animal Experimentation and Care (NOM-062-ZOO-2001) and the guidelines outlined by the American Society of Ichthyologists and Herpetologists (American Society of Ichthyologists and Herpetologists (ASIH), 2004).

Warming trials

We conducted snake warming trials during the period of highest monthly activity (May to November, Manjarrez, 1998) using semi-natural enclosures, constructed with plastic mesh of 6.3 m2 (2.8 × 2.4 m) and a height of 0.7 m. One enclosure was placed in the forest (trying to have sun and shade), and the other was placed in the grassland (Fig. 1). The enclosures kept in the same location the entire May to November activity season. Free-ranging Thamnophis spp. could be implicated in activities that conflict with thermoregulation (Gregory, 1984), however in enclosures gartersnakes show enhanced thermoregulatory control, they are able to regulate Tb within a narrow range defined as the plateau phase (sensu Peterson, 1987). This was the most common pattern observed (Peterson, 1987; Giacometti et al., 2021).

We attempted to recreate the thermal environmental conditions of both environments (mean substrate temperature: forest = 20.49 ± 4.4 °C; grassland = 21.65 ± 3.9 °C). At the same time, three adult males (>395 mm snout-vent length SVL; Manjarrez, 1998) were introduced to each enclosure (sexed by visual inspection of tail-base breadth and measured in SVL with a tape measure, mean ± SD, 503.5 ± 73.4 mm; body mass with a Pesola 41.24 ± 9.70 g) and three adult females (SVL, mean ± SD, 544.2 ± 94.0 mm; body mass 54.60 ± 1.09 g), which were fed one day before the test. Different individuals were used each month, and each individual was studied in a single type of habitat. The snakes were introduced at 07:00 h to acclimate to the environment under natural sunlight, and from 08:00 to 12:30 h, we recorded the cloacal, substrate and air temperature (as previously described) every 30 min, using a Schultheis thermometer, maintaining individual monitoring. The snake temperatures were measured in both enclosures on the same days and at the same time. Although care was taken to avoid frightening the snakes, they were very sensitive to the presence of the recorder. Unintentional disturbances occurred on approximately 10% of the records. Data from disturbed snakes were discarded.

Analyses

In the warming trials, each snake was exposed to only one environment (grassland or forest; Weatherhead et al., 2012) and then measured Tb, Ts and Ta at 30-min intervals. We performed a repeated measures two-way ANOVA (Gotelli & Ellison, 2004) to determine whether the mean Tb in each temporal interval (ten intervals; 30 min from 08:00 to 12:30 h) differed significantly according to two sources of variation: month (6 months; May to November, grouped into three seasons) and environment (grassland and forest). Comparisons of Ta and Ts between seasons were made with ANOVA or Kruskal–Wallis, depending on the case. We employed repeated measures ANOVA and used multivariate Wilk’s lambda test because the assumptions of compound symmetry and sphericity rarely hold (Taylor, DeNardo & Malawy, 2004). The null hypotheses were as follows: (a) the Tbs between seasons were statistically similar, (b) the Tbs between environments were statistically similar, and (c) the Tbs did not differ depending on the season-environment interaction where the snakes were recorded.

The differences between the environment and seasons were analyzed with an Unequal N HSD post hoc test to estimate differences between groups via main factor analysis and a bifactorial analysis at 10:30-min intervals. This post hoc test is a modification of the Tukey HSD test and was used to determine the significance between group means with balanced sample sizes values. The MANOVA post hoc test was performed for each 30-min interval (10 intervals from 08:00 to 12:30 h); therefore, we estimated differences among environments and seasons at the same temporal interval. In the warming trials, Ts was compared using Mann–Whitney U test. We grouped males and females because separate analyses indicated that both sexes exhibited similar patterns of Tb (P = 0.23) Snake size also had no significant effect on Tb pattern (P = 0.17). We conducted MANOVA tests with Statistica 13.0 software. Alpha was set at 0.05 for all tests. The data are reported as the mean ± 1SD.

Results

In the warming trials, Ts in the grassland (22.3 ± 3.9 °C, n = 447 records) was greater than that in the forest (19.6 ± 5.0 °C, n = 447 records, Mann–Whitney U test = 5,154, P < 0.0001, z-adjusted = −8.4, P < 0.0001; Fig. 2). In the forest, Ta was higher in spring and summer than in autumn (K-W = 11.1, P < 0.0001; Table 1; Fig. 2). Ts was higher in spring and summer than in autumn (K-W = 6.2, P < 0.04; Table 1). Ts remained constant from 08:00 to 09:30 h (11.4 °C) and increased from 10:00 to 11:00 h (from 14.46 °C to 19.4 °C), to stabilize again from 11:30 to 12:30 h (22.3 °C, F9,60 = 12.24, P < 0.0001). Coinciding with the body warming of the snakes (Fig. 3). In the grassland, Ta increased steadily until 10:30 h (from 11.6 °C to 19 °C, F9,69 = 15.3, P < 0.0001), then stabilized around 20.5 °C (Fig. 2). The mean Ta in spring and summer was higher than in autumn (K-W = 11.16, P < 0.003; Table 1). Ts in spring and summer was 8.2 °C higher than in autumn (F2,69 = 44.5, P < 0.0001), and generally stabilized at 10:00, around 23 °C, coinciding with the time when snakes reach the plateau phase.

Figure 2 (A–G) Diurnal variation in air and substrate temperature.

Mean temperatures (°C) registered each 30-min within semi-natural enclosures constructed in forest (blue solid line) and grassland (red dotted line) from May to November. Air temperature (Ta), registered at 1.5 m above the ground, and substrate temperature (Ts), 2 cm above ground.

Table 1 Seasonal variation in mean ambient (Ts, Ta) and body temperature (Tb) of T. eques in the forest and grassland during warming trials.

	°C (± SD)	
Season	Grassland	Forest	
	Tb	Ta	Ts	n	Tb	Ta	Ts	n	
Spring	30.9 ± 4.8	21.3 ± 4.4	24.4 ± 5.4	54	22.4 ± 9.0	18.4 ± 3.8	20.3 ± 6.2	51	
Summer	24.1 ± 5.9	17.0 ± 2.8	19.1 ± 3.5	141	19.7 ± 6.0	16.3 ± 2.6	17.2 ± 3.2	144	
Autumn	21.5 ± 8.4	15.0 ± 4.8	16.2 ± 5.2	142	20.0 ± 8.8	13.4 ± 4.6	14.7 ± 5.7	159	

Figure 3 Diurnal variation in body temperature (Tb) of T. eques.

Mean Tb (°C ± SD) recorded each 30-min within semi-natural enclosures constructed in forest (blue solid line) and grassland (red dotted line) from May to November obtained from MANOVA.

We found that the Tb of T. eques varied significantly between environments (Wilks λ = 0.53, F10,45 = 3.9, P < 0.0001), seasons (Wilks λ = 0.0001, F60,240 = 17.7, P < 0.0001), and interactions of both factors (Wilks λ = 0.001, F60,240 = 9.8, P < 0.0001). The average Tb in the grassland was 0.88 °C greater than that in the forest, while the Tb in October (21.5 °C) and November (17.5 °C) was the lowest, and that in May (26.3 °C) and July (23.8 °C) was greater. The diurnal body heating in both environments showed a typical daily thermoregulation pattern, with a rapid increase in the morning (08:00–10:30 h), followed by a stabilized phase at noon (11:00–12:30, Fig. 3). The main factor analysis of the environment revealed that the mean Tb of grassland was significantly greater than of forest at 09:30, 10:30–11:00, and from 12:00 to 12:30 (difference in Tb between the environments: 1.6 °C, 0.8 °C, 1.9 °C, 1.3 °C, and 1.5 °C, respectively), while at 08:00 and 08:30, the mean Tb of forest was greater than that of grassland (difference in Tb between the environments: 0.94 °C and 0.4 °C, respectively; Wilks λ = 0.41, F 0,80 = 6.9, P < 0.0001; Fig. 3). The environments influenced the body warming of the snakes, because in the grassland the snakes reached the stable phase Tb (28.7 °C, F9,355 = 117.6, P < 0.0001) between 10:00 and 10:30, while in the forest it was between 11:00 and 11:30 (27.2 °C), one hour later. The Tb of the stable phase of the grassland was located on average 1.5 °C above the forest. We did not find significant differences in Tb between forest and grassland at 09:00, 10:00 and 11:30 h (P = 0.38, 0.07 and 0.25, respectively). The variation of Tb throughout the day was correlated with the Ta and Ts, both in the grassland and in the forest, a pattern that was constant during all months and seasons (Table 2).

Table 2 Pearson correlation coefficients of T. eques Tb with Ta and Ts, in grassland and forest, during spring, summer and autumn months.

Season	Grassland	Forest	
	Tb-Ta	Tb-Ts	n	Tb-Ta	Tb-Ts	n	
Spring	68.3	73.4	53	88.1	88.9	51	
Summer	84.4	82.7	139	81.5	85.4	144	
Autumn	86.5	87.0	142	62.0	78.2	159	
Note:

P-values ≤ 0.0001 in all tests.

A lower number of differences in Tb among seasons and environments at the same temporal interval was recorded in the late rise phase and early plateau phase (10:00–11:30 h); an intermediate number of differences occurred in the late plateau phase (12:00–12:30 h); and a greater number of differences were recorded in the early rise phase (08:00–09:30 h, Fig. 4). We found that most of the differences in Tb among environments at the same month (season) and temporal interval were recorded in the early rising phase in late spring and in early summer, according to our prediction, when tree canopy cover was continuous (May–September). We found that significant differences between grassland and forest were recorded from 08:00 to 09:30 in May, June, and July, whereas Tb was similar from 10:00 to 12:30 in both environments. In August, September, October, and November, we did not find significant differences in 23 of the 28 comparisons of Tb among the grassland and forest from 08:00 to 12:30 (Fig. 4). These latter results partially support our prediction of Tb equality (grassland vs forest) when tree canopy cover is absent but do not exactly coincide with the months of absence of leaves (October–November).

Figure 4 Number of post hoc significant differences derived from MANOVA.

The MANOVA post hoc test was performed analyzing Tb of T. eques for each 30-min interval (10 intervals from 08:00 to 12:30 h), within grassland (gray bars), within forest (white bars) and between both environments (black bars).

Discussion

In this study, we analyzed the daily and seasonal Tb of T. eques in forest and grassland of a Mexican locality. According to our prediction, our results show that Tb differences between grasslands and forests occurred principally in late spring and early summer during the early rise and late plateau phases, coinciding with the presence of foliage on the willow trees present in the study area, while Tb was similar in late summer and early autumn, when tree canopy cover is absent. Seasonal and environmental variations in Tb snakes have been widely documented in template zones (Harvey & Weatherhead, 2010; George, Connette & Thompson FR, 2017; Johnson, Poulin & Somers, 2022; Figueroa-Huitrón et al., 2024b). Milk snakes (Lampropeltis triangulum) thermoregulate more effectively in the spring than in the summer and fall and more effectively in forests than in open habitats (Row & Blouin-Demers, 2006), as has also been reported for the black rat snake Elaphe obsoleta (Blouin-Demers & Weatherhead, 2001a, 2002). In Arizona, habitat selection of T. eques varied by season (Sprague & Bateman, 2018). During the active season (March–October), it primarily selected wetland edge habitats with abundant vegetative cover. Gestating females selected similar locations but with less dense vegetative cover. In our study, we propose that the Tb of T. eques in both forest and grassland was similar when tree canopy cover was absent (late October to November) and differed when tree canopy cover was continuous. This is probably explained because in fall, when tree canopy cover is absent increases the amount of solar radiation on the substrate and an open habitat is formed with constant direct sunlight (Weatherhead et al., 2012; Lara-Reséndiz et al., 2014). Our results suggest that temporal variations in environmental thermal properties affect the Tb of snakes. We report how temporal changes in Tb between two different environments tend to converge when canopy cover is similar, in this case, when tree canopy cover is absent. This possible convergence in Tb of T. eques during the opening of sunny areas could also be a response of individuals who choose open areas to favor the embryonic development rates of pregnant females (Sprague & Bateman, 2018), however, during this season, there are few pregnant females T. eques, because the gestation time in this population occurs mainly during spring and summer (Manjarrez & San-Roman-Apolonio, 2015). The responses of snakes to these temporal and spatial thermal changes could influence their proximal and ultimate strategies, which affect their life-history traits and fitness (O’Donnell & Arnold, 2005; Gangloff, Vleck & Bronikowski, 2015).

The lower number of differences at the late rise and early plateau phases among seasons (months) and environments could be explained by the greater variations in Tb at these phases. At these temporal intervals, different individual behavioral strategies could affect their Tb and increase their variability (Peterson, Gibson & Dorcas, 1993), in contrast to early rise and late plateau phases, where individuals tend to experience similar behavioral mechanisms and therefore lower variability. A single behavioral mechanism does not necessarily dominate a given area, and several mechanisms can coexist if they entail similar fitness gains (Leliévre et al., 2010). The proximal repercussions of these behavioral mechanisms, such as time spent basking or under rocks, could affect the time inverted on other activities that maximize their fitness, such as feeding, growing or reproduction (Vitt & Caldwell, 2009; Gangloff, Vleck & Bronikowski, 2015). For example, juvenile broad-headed snakes (Hoplocephalus bungaroides) rarely bask or move by day, whereas some adults are active by day and bask in the open arena, possibly to reduce avian depredation (Webb & Shine, 1998). In snakes, temporal and spatial variations in Tb represent the behavioral flexibility of individual animals rather than genetic differences between populations (Shine, 1987; Arnold, Peterson & Gladstone, 1995). However, it is necessary to estimate the ultimate repercussions on survival and reproduction as a result of individual behavioral mechanisms followed by snakes in the late rise and early plateau phases (Vitt & Caldwell, 2009).

Thamnophis eques is a southern Gartersnake species with broader geographic distributions that occur under a variety of environmental conditions, from deciduous scrubland to pine-oak forest (Rossman, Ford & Seigel, 1996). Rosen (1991) reported that T. eques is the Gartersnake with the lowest Tb (28 °C) in a snake community in Arizona, United States. He noted that T. eques prefers cooler environments despite the availability of warmer areas. However, this result may not be valid for the entire wide distribution and consequent diversity of habitats of T. eques. Also in Arizona, during the active season of T. eques (March–October), Sprague & Bateman (2018) report snake Tb, recorded from transmitters, of 29.3 °C for females and 27.5 °C for males. Conant (2003), on Lake Cuitzeo (Michoacan, Mexico), reported a Tb near 28 °C in snakes found hiding in rockpiles. We found that the Tb at the plateau phase in two different thermal environments was nearly 28 °C. Arizona populations represent a boreal limit of distribution of Mexican Gartersnake, whereas Lake Cuitzeo and this study population (at Toluca) from Central México, occur in the middle of their geographic distribution (Rossman, Ford & Seigel, 1996). Therefore, a Tb near 28 °C in the plateau phase could be a constant characteristic across the distribution of this southern Gartersnake. Our results support a conservative view, and they do not vary greatly intraspecifically (Angilletta, Niewiarowski & Navas, 2002), at least when both environments are thermally contrasting, before the leaves fall. Its semiaquatic lifestyle may aid it to conserve beneficial Tb with less effort since aquatic environments are more thermally homogeneous, as suggested for the sympatric species T. melanogaster (Figueroa-Huitrón et al., 2024b). This pattern is also present in T. sirtalis and T. elegans (Gibson & Falls, 1979; Lillywhite, 1987; Peterson, Gibson & Dorcas, 1993), which maintain selected temperatures in the field from 27 °C to 33 °C, and in lizard species such as Sceloporus, with a mean Tb of 35 °C throughout their elevational range at temperate latitudes (Andrews, 1998). Although some research has shown that the thermal traits of some snakes vary depending on habitat type (Row & Blouin-Demers, 2006; Harvey & Weatherhead, 2010; Jaramillo-Alba et al., 2020; Raya-García & Arteaga-Tinoco, 2024), indicating that these traits are more plastic than previously suggested (Andrews, 1998). On the other hand, it has also been suggested that phylogeny explains most of the variation in Tb and upper thermal limits, at least for lizards (Grigg & Buckley, 2013), the group with the most thermal studies in reptiles. We encourage the development of thermoregulatory studies on T. eques throughout its entire geographic distribution to test whether Tb is spatially constant.

Point-sampling studies have been severely criticized because these records are highly sensitive to deviations from true randomness; however, point sampling tends to have seasonal trends similar to those of semicontinuous sampling (Taylor, DeNardo & Malawy, 2004). We found similar seasonal variations in Tb to those obtained using semicontinuous sampling (Shine & Lambeck, 1990; Taylor, DeNardo & Malawy, 2004; Plummer & Mills, 2010); therefore, our results could represent the real spatial and temporal variations in T. eques Tb. We cautioned that the results presented here are applicable only to temporal intervals that were evaluated and encouraged for exploring the use of semicontinuous sampling to improve estimates of temporal and spatial variations in Tb. Besides, the study of the thermal ecology of snakes under semi-natural enclosures is particularly important for a more realistic understanding of thermal responses and their impact on habitat selection (Isaac & Gregory, 2004; Noble, Carazo & Whiting, 2012). Such studies are usually replaced by laboratory studies, under conditions that fail to reflect the type of ecological problems faced in the wild (Peterson, 1987). Semi-natural enclosures allow snakes access to the full range of thermal cues that they would have access to in their natural habitat, whereas laboratory experiments typically focus on a restricted range of thermal cues. In our enclosure, the trade-offs between risks and benefits presumably are involved in decisions to adopt thermoregulatory behavior of T. eques (Todd, Jodrey & Stahlschmidt, 2016; Herr et al., 2020).

Mexican Gartersnake is a semiaquatic species that requires a combination of terrestrial and aquatic resources (Rossman, Ford & Seigel, 1996; Manjarrez, 1998; Emmons, Nowak & Lauger, 2016; Manjarrez, Pacheco-Tinoco & Venegas-Barrera, 2017). In our study location, grasslands and forests offer different environmental conditions that permit the local persistence of T. eques. The forest surrounds the waterbody and offers a greater availability of terrestrial and aquatic preys (Manjarrez, 1998; Manjarrez, Pacheco-Tinoco & Venegas-Barrera, 2017), and a more constant temperature environment than does the grassland. On the other hand, in the grassland, there is a greater availability of ground retreats as burrows (CS Venegas-Barrera, 1999, personal communication) and reaches the plateau phase early. Burrows provide thermal heterogeneity because they offer snakes stable environmental temperatures even when surface environmental temperatures are potentially lethal (Johnson, Poulin & Somers, 2022; Raya-García & Arteaga-Tinoco, 2024; Figueroa-Huitrón et al., 2024b). Therefore, we propose that both environments are necessary for the local and regional persistence of T. eques (Sprague & Bateman, 2018). The current fragmented distribution of Mexican Gartersnakes in Mexico appears to be the result of anthropogenic destruction of its habitat. In Mexico, from 1993 to 2000, an annual deforestation rate of 1.02%, a loss of natural grasslands of 1.97% and an increase in cropland of 1.96% were estimated (Velázquez et al., 2002); in 2001–2018, an average of 212,070 ha were lost per year (CONAFOR, 2020). Today, T. eques inhabits fragmented and degraded habitat in Central Mexico (Rossman, Ford & Seigel, 1996; Conant, 2003). Such a scenario could reduce the environmental conditions for the abundance, reproduction and survival of this Gartersnake species associated with water bodies that require a mixed combination of environments (Center for Biological Diversity, 2003; Kapfer, Doehler & Hay, 2013). In a snake community in American tallgrass prairies, it has been suggested that the overall decline in the abundance of snakes is associated with an agricultural matrix that may limit snake populations via mortality or decreased dispersal abilities (Cage, 2008). Several species require a combination of different environments to complete their life cycle (Reinert, 1993; Law & Dickman, 1998), which offers different food availability, retreats or thermal conditions that favor their fitness. For example, black rat snakes (Elaphe obsoleta), which have the lowest thermal quality in open habitats (higher retreat sites) compared with edge and forest habitats, experienced more favorable body temperatures in open habitats than in edge and forest habitats (Blouin-Demers & Weatherhead, 2002), a pattern similar to that presented by Coluber constrictor in his preferences for forest edges in Ontario and Illinois (Carfagno & Weatherhead, 2006). Reproductive conditions and the thermal environment also affect snake habitat selection; for example, gravid females of T. eques and other snakes like Timber rattlesnake (Crotalus horridus, Reinert & Zappalorti, 1988), viper snakes (Vipera aspis, Muri et al., 2015), and black rat snakes (Elaphe obsoleta obsoleta, Blouin-Demers & Weatherhead, 2001c), are more strongly associated with open or edge habitats than are nongravid females and males (Reinert & Zappalorti, 1988; Blouin-Demers & Weatherhead, 2001c; Sprague & Bateman, 2018). In semiaquatic snakes, the presence of terrestrial habitats around wetlands may be crucial for surviving drought (Willson et al., 2006; Kapfer, Doehler & Hay, 2013; Figueroa-Huitrón et al., 2024b). We propose the development of a radiotelemetry study to estimate temporal variations in the dispersion of T. eques between environments related to temporal changes in environmental temperatures (Sprague & Bateman, 2018).

Conclusions

In conclusion, our results support the hypothesis that the Tb of T. eques diverged between grasslands and forests when willows presented leaves and was similar when the canopy cover was similar, in this case, when tree canopy cover was absent. Our results suggest that the daily and seasonal warming patterns of T. eques are similar to those of other northern Gartersnakes (Peterson, Gibson & Dorcas, 1993; Rosen, 1991; Rossman, Ford & Seigel, 1996). Thamnophis eques is considered a threatened species (SEMARNAT, 2010; Fish and Wildlife Service, 2023); however, in Mexico, where it has its widest distribution, there is a lack of information on the ecology and current conservation status of its populations in the different habitats (González-Fernández et al., 2018). Therefore, it is necessary to consider appropriate management strategies to achieve effective conservation of this species.

Supplemental Information

Supplemental Information 1 Data temperatures.

The raw data show the records obtained for snakes during all surveys. These data were used for the figures and statistical analysis.

Supplemental Information 2 Data repeated ANOVA.

We express our sincere appreciation to Erika Nowak and three anonymous reviewers for their valuable feedback, which greatly contributed to improving the quality of this manuscript.

Additional Information and Declarations

Competing Interests

Author Contributions

Animal Ethics

Ethics

Data Availability

Javier Manjarrez and Armando Sunny are Academic Editors for PeerJ.

Crystian S. Venegas-Barrera conceived and designed the experiments, performed the experiments, analyzed the data, prepared figures and/or tables, authored or reviewed drafts of the article, and approved the final draft.

Armando Sunny analyzed the data, authored or reviewed drafts of the article, and approved the final draft.

Javier Manjarrez conceived and designed the experiments, prepared figures and/or tables, authored or reviewed drafts of the article, and approved the final draft.

The following information was supplied relating to ethical approvals (i.e., approving body and any reference numbers):

This study received the approval of field permit (Secretaria del Medio Ambiente y Recursos Naturales #07164) and the ethics committee of the Universidad Autónoma del Estado de México (Numbers 3589/2013SF, 4047/2016SF, and 4865/2019SF).

The following information was supplied relating to ethical approvals (i.e., approving body and any reference numbers):

Secretaria del Medio Ambiente y Recursos Naturales and the Universidad Autónoma del Estado de México

The following information was supplied regarding data availability:

The raw measurements are available in the Supplemental File.

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
