# Peer review of "Thermal ecology of the Mexican Garter Snake (Thamnophis eques): temporal and spatial variations"

_PeerJ, doi:10.7717/peerj.18641_

## Round 0.1 · original submission · Major Revisions

Please revise your manuscript, making changes to address all of the reviewers' comments. Include a document describing all of the changes.

Reviewer 1 ·

Basic reporting

I reviewed this manuscript entitled “Thermal ecology of Mexican garter snake: temporal and spatial variations” (104845-v0), which could be potential to be published in this PeerJ journal, as long as authors make significant adjustments in all sections. The ecological and evolutionary terms indicated in the work related to the thermal ecology of the snake Thamnophis eques are weak, so the authors should review the literature on the subject very carefully. Here I make some observations, but in the text, they are all there.

Experimental design

8. Methods- Lines 113-115- How much is the variation in environmental temperatures, for example, mean temperature in summer 22 °C to 25°C in fall, could you explain it a little bit to understand the environmental changes in temperature?, please? Because in this context we can follow the behavior of the snake face to the environmental
9. Lines 140-141- Could you be more specific, please, since you have not yet spoken about the environmental conditions of both sites, and this is necessary to see how the snakes respond.
10. Line 142- What about the weight (body mass), it is very important in thermoregulation of ectothermic organisms. On the other hand, it is important that the methods section be supported by diverse and more recent references on thermal ecology in snakes, there is sufficient literature.
11. Analysis- lines 148- 169- See comments on text,
12. In all methods section, I didn´t see the importance of Tb and Ta on body temperature of the snake, and it is very important in ectothermic reptiles (lizards and snakes), please explain why there is no regressions analysis?

Validity of the findings

13. Results section- Lines 172-175- for example, n= 447, what does it mean?, records of time?, temperature of substrate?, please be clearer.
14. Lines 177-179. But my question remains; what is the sample size, or are they repetitions of taking temperatures from a single individual?
15. Lines 182-187- But what about Ta and Tm, at some times of the day body temperature correlates with Ta and Tm, and at other times it does not?
16. Discussion section- Line 204- see comments on text, because the seasonal term was not clear in results section.
17. Lines 207-209- So, the body temperatures of the snakes were similar in this period, meaning that in this way "similar in body temperatures", they respond to low temperatures (Tb?), as the ambient temperatures decrease; could be clearer, please!
18. Lines 216-218- cause-effect, causes changes in the environment (falling leaves from trees), effect, similar thermoregulation of snakes in both environments, so, biologically, why do they respond like this (similar)? Please explain!
19. Line 220- about it, If in this cold period, the open area favors them to thermoregulate in a similar way in both environments, it could be a response of the individuals of the snake populations, which being ectothermic, viviparous, and possibly gestating, the open areas favor them to have a rate of embryonic development in which it does not vary so much, this could be a response, something indicates this similar thermoregulation in both places.
20. Lines 252-253- This may be true, since in oviparous and viviparous lizard species, these species, physiologically retain (conservatism) a narrow range among populations of the same species, but they must explain the cause of conservatism, since if in one environment and another they differ in spring and summer, then, conservatism does not apply in this case.
21. Lines 275-276- Yes, but according to the results, the biological and ecological characteristics of both sites are not indicated, and this must be clear to understand the thermoregulatory responses of the species
22. 282-299- I did not find any record of optimal temperatures for thermoregulation, this is vital, because if climate change is affecting its thermoregulation range, then it could be changing the history of its conservatism; with the information indicated, and the record by other authors of the body temperature of 28 ° C in T. eque, I would assume that its thermoregulation range is 17.5 ° C to 28 ° C, which is what is indicated here, but its optimal temperature is not indicated, this is important because it has to do with all its activities, such as foraging, escape from predators, reproduction, gestation, among others.

Additional comments

Other comments are inside of manuscript, but here add all the comments indicated above:
Comments on the manuscript

I reviewed this manuscript entitled “Thermal ecology of Mexican garter snake: temporal and spatial variations” (104845-v0), which could be potential to be published in this PeerJ journal, as long as authors make significant adjustments in all sections. The ecological and evolutionary terms indicated in the work related to the thermal ecology of the snake Thamnophis eques are weak, so the authors should review the literature on the subject very carefully. Here I make some observations, but in the text, they are all there.
1. Title should be changed to “Thermal ecology of Mexican garter snake, Thamnophis eques: temporal and spatial variation”, something like that, because the words “Mexican garter snake”, It means that they are treating all species of the Thamnophis genus from Mexico, and this is not the case.
2. Introduction- Lines 38-39, this is a general idea on the term “thermal environment”, this apply for all animal (invertebrates and vertebrates), so could talk in general and after that, specifically on snakes.
3. Lines 43-44- “Thermal patches…” these kind of environments or microenvironments are chosen by individuals to initiate multiple activities, such as foraging, escaping predators, among others.
4. Lines 48-49- the idea should be related to the ectothermic and daily activity of snakes, see comments on text.
5. Lines 61-62- are you talking about ontogenetic changes in habitat selection?
6. Lines 72-75- Regarding this idea, you could propose a hypothesis about the thermoregulation of the snake in a gradient with different types of vegetation, it is a suggestion!
7. Lines 100-105- And what about the temperature conditions of the environment, such as air, substrate, since this species of snake, as ectothermic, depends on the conditions of the environment, and here the Tb is isolated from the environment.
8. Methods- Lines 113-115- How much is the variation in environmental temperatures, for example, mean temperature in summer 22 °C to 25°C in fall, could you explain it a little bit to understand the environmental changes in temperature?, please? Because in this context we can follow the behavior of the snake face to the environmental
9. Lines 140-141- Could you be more specific, please, since you have not yet spoken about the environmental conditions of both sites, and this is necessary to see how the snakes respond.
10. Line 142- What about the weight (body mass), it is very important in thermoregulation of ectothermic organisms. On the other hand, it is important that the methods section be supported by diverse and more recent references on thermal ecology in snakes, there is sufficient literature.
11. Analysis- lines 148- 169- See comments on text,
12. In all methods section, I didn´t see the importance of Tb and Ta on body temperature of the snake, and it is very important in ectothermic reptiles (lizards and snakes), please explain why there is no regressions analysis?
13. Results section- Lines 172-175- for example, n= 447, what does it mean?, records of time?, temperature of substrate?, please be clearer.
14. Lines 177-179. But my question remains; what is the sample size, or are they repetitions of taking temperatures from a single individual?
15. Lines 182-187- But what about Ta and Tm, at some times of the day body temperature correlates with Ta and Tm, and at other times it does not?
16. Discussion section- Line 204- see comments on text, because the seasonal term was not clear in results section.
17. Lines 207-209- So, the body temperatures of the snakes were similar in this period, meaning that in this way "similar in body temperatures", they respond to low temperatures (Tb?), as the ambient temperatures decrease; could be clearer, please!
18. Lines 216-218- cause-effect, causes changes in the environment (falling leaves from trees), effect, similar thermoregulation of snakes in both environments, so, biologically, why do they respond like this (similar)? Please explain!
19. Line 220- about it, If in this cold period, the open area favors them to thermoregulate in a similar way in both environments, it could be a response of the individuals of the snake populations, which being ectothermic, viviparous, and possibly gestating, the open areas favor them to have a rate of embryonic development in which it does not vary so much, this could be a response, something indicates this similar thermoregulation in both places.
20. Lines 252-253- This may be true, since in oviparous and viviparous lizard species, these species, physiologically retain (conservatism) a narrow range among populations of the same species, but they must explain the cause of conservatism, since if in one environment and another they differ in spring and summer, then, conservatism does not apply in this case.
21. Lines 275-276- Yes, but according to the results, the biological and ecological characteristics of both sites are not indicated, and this must be clear to understand the thermoregulatory responses of the species
22. 282-299- I did not find any record of optimal temperatures for thermoregulation, this is vital, because if climate change is affecting its thermoregulation range, then it could be changing the history of its conservatism; with the information indicated, and the record by other authors of the body temperature of 28 ° C in T. eque, I would assume that its thermoregulation range is 17.5 ° C to 28 ° C, which is what is indicated here, but its optimal temperature is not indicated, this is important because it has to do with all its activities, such as foraging, escape from predators, reproduction, gestation, among others.

Annotated reviews are not available for download in order to protect the identity of reviewers who chose to remain anonymous.

Reviewer 2 ·

Basic reporting

There are multiple instances where revisions for grammar and clarity are needed.
Literature references, background, and context are sufficient.
Figures 2, 3, and 5 might be omitted.
Because a repeated-measures design was used, the supplemental data file should also include a variable providing a snake ID indicator. Also, the ground surface temperature data is not provided.
I am confused by hypothesis c – does it refer to a month by environment interaction?

Experimental design

This manuscript meets the aims and scope of PEERJ in that it is "in the Biological Sciences, Environmental Sciences, Medical Sciences, and Health Sciences." I couldn't find anything more specific regarding PEERJ Aims and Scope.
The authors aim to fill the knowledge gap regarding “how temporal variations in the structure of vegetation at local scales can affect daily snake thermoregulation patterns.”
Comments on the rigor and methods are provided in the Additional Comments below.

Validity of the findings

Because a repeated-measures design was used, the supplemental data file should also include a variable providing a snake ID indicator. I expected the total number of entries in the supplemental data file to be 840 (6 snakes X 10 times X 7 months X 2 environments). Instead, there are 791 entries.
Also, the ground surface temperature data is not provided.
I am a little rusty on repeated measures analyses but typically they include within-subjects factors (time and season in this study), between-subjects factors (environment in this study), and interactions (here, I think those would within-subjects interactions of time-by-environment, month-by-environment, time-by-month, and time-by-month-by-environment). [It seems that perhaps time was not included as a source of variation in the analyses presented in the manuscript.] In addition, one has a choice of contrast types (deviation, simple, Helmert, repeated, polynomial) among factor levels. Neither the full array of interactions, nor the contrast type used, are reported in the manuscript. And given the number of sources of variation, my preference would be to have the statistical results presented in the form of a table.
Conclusions should more directly address the month-by-environment interaction and not the main effects of month and environment.

Additional comments

Line 28 and elsewhere: I suggest that “presented” be replaced with “showed” or “exhibited”
Line 55: change to “Thermal studies have shown . . .”
Line 57: I suggest that “noon” be replaced with “mid-day”
Line 65: I suggest changing “The garter snake species (Thamnophis) is . . .” to "Garter snake species (Thamnophis) are . . .”
Line 68-72: break this into two sentences
Line 74: clarify or omit the phrase “making it a biological model to analyze variations in their thermoregulation on contiguous environment”
Line 81-82: edit for clarity; what is meant by “warmer cooler” and “attenuated and stable”?
Line 83: I suggest editing “A reduction in the canopy” to make clear that you mean the seasonal loss of leaves from deciduous trees
Line 84-86: This is the third time garter snakes have been described as a “model” – that point has been made!
Line 87-92: The sentence seems to contrast the thermal conditions of grassland and forest habitats but both are described as heterogeneous – I’m confused.
Line 92: replace “become forbidden” with a clearer phrase
Line 96-97: change “throughout daily and seasonally during the free-ranging period of activity” to something like “throughout daily and seasonal periods of activity”
Line 100: replace “scenarios” – do you mean “hypotheses”?
Line 111: by “floodable” do you mean “seasonally flooded” or perhaps “floodplain”?
Line 113: omit “Both”
Line 116-118: move this to the paragraph at lines 128-133
Line 118-126: This information doesn’t really belong in a paragraph called Study Area
Line 124-126: possible revision - “From March to November 1999, we characterized grassland and forest thermal environments by recording ground surface temperature every five minutes from 1030 to 1230 h (Ts ±0.2 °C) at random points using a Schultheis quick registering thermometer.”
Line 136: Change “To analyze temporal variations in the Tb between environments, during the period of highest monthly activity (May to November, Manjarrez, 1998), snake body warming was performed in a seminatural experiment,” to “We conducted snake warming trials during the period of highest monthly activity (May to November, Manjarrez, 1998) using semi-natural enclosures.” Then start a new sentence describing the enclosures.
• Were the enclosures kept in the same location the entire May to November activity season or moved from trial to trial?
• What was done to recreate the thermal environmental conditions of both environments?
• From the figures, it looks like these experiments were done once a month - is that correct?
• Were all six snakes placed in enclosures at the same time?
Line 135-145: The enclosures appear to be located more than a kilometer apart. Were snake temperatures measured in both enclosures on the same days and at the same time? If so, how was this accomplished? If not, how were measurement scheduled to allow for meaningful comparisons?
Fig. 1. Should the labels “Grassland enclosed” and “Forest enclosed” be “Grassland enclosure” and “Forest enclosure”; do the photos actually show the enclosures? If so, I don't see them even when I enlarge the figure.
Line 141: Information on snake size is somewhat confusing. What is D.E. an abbreviation for? I suggest providing the range in snake size as well.
Line 143: Were cloacal temperatures also measured with a Schultheis quick registering thermometer?
Line 149-151: Omit this sentence and begin the next sentence with “We”
Line 151-154 and supplemental data: Because a repeated-measures design was used, the supplemental data file should also include a variable providing a snake ID indicator. Also, the ground surface temperature data is not provided.
Line 154-156: Simplify - “We employed repeated measures ANOVA and used multivariate Wilk’s lambda test because the assumptions of compound symmetry and sphericity rarely hold (Taylor, et al., 2004).” As noted earlier, the statistical analyses need fuller explanation.
Line 156-158: Doesn't this repeat the information in lines 152-154?
Line 165: Why the italicization?
Line 160-161: I am confused by hypothesis c – does it refer to a month-by-environment interaction?
Line 172-174: these refer to temperatures from random points, correct? It seems like they do not belong in a paragraph labeled “Warming trials.” Can you say anything about how the difference between habitats varied seasonally (this might help explain the Tb results)?
Line 179: move the temperatures (in parentheses) adjacent to the month named
Lines 174: Can information be provided to indicate that snakes in enclosures actively thermoregulated in a way similar to free-ranging snakes? Snakes are being placed into an unfamiliar enclosure whereas free-ranging snakes likely have familiarity with their spatio-temporal thermal environment and use that familiarity to modify Tb.
Lines 174-201: given that there is a significant time-by-month interaction, the main effects of time and month are not particularly interesting. I suggest emphasizing the characterization of that interaction in the Results. It is well illustrated in Figure 4; I think Figure 2 and 3 could be omitted. Post-hoc analyses, if included at all, should be used to compare time-month combinations and levels of the main effects. However, could argue that time and month are both random effects and so variance components might be more appropriate than post-hoc comparisons.
Line 203: Somewhere in the discussion, the advantages and disadvantages of using semi-natural enclosures should be evaluated.
Line 203: Given the hypothesized impact of leaf fall on thermal environments, it would have been helpful if dates of leaf fall were recorded along with measures of the amount of light reaching the enclosures.

Reviewer 3 ·

Basic reporting

I have reviewed the work titled “Thermal ecology of Mexican garter snake: temporal and spatial variations” submitted to the journal PeerJ. In general, the manuscript is well written, but basic methodological aspects are lacking. Therefore, it is necessary to stablish clearly and directly these aspects in the manuscript. I have also suggested adding up-to-date references to complement those found in the manuscript.

Experimental design

data has been analyzed properly; however, I have made some suggestions to authors in order to improve results and conclusions of the work.

Validity of the findings

I consider that this is a great and interesting contribution to thermal ecology of Mexican snakes (especially Thamnophis species).

Additional comments

I have uploaded a PDF with specific suggestions arranged by line number throughout the manuscript, hoping that authors consider my recommendations to make this version more publishable.

Annotated reviews are not available for download in order to protect the identity of reviewers who chose to remain anonymous.

·

Basic reporting

Basic Reporting: Overall, the manuscript has a professional style, with all necessary elements included and well-thought out. The hypotheses are clearly stated and are generally well-addressed in the results.

The manuscript would benefit from an additional read-over by a native English-speaking author; some of the writing is not quickly interpretable or awkward due to slightly unexpected phrasing, e.g. “when willows present leaves”. In particular, the abstract could be re-written to be more concise and clearer.

The introduction, discussion, and conclusions suffer from lack of consideration of habitat studies on T. eques in the U.S.; these sections are incomplete. The figures could be improved by adding more detail to figure legends, and some figures would benefit from adding more data. The raw data is present, but was not well annotated, so I’m a little unclear how it was summarized for analyses.

Experimental design

The hypotheses are well-defined, and the research question seems potentially useful to species conservation. The methods need additional details added so that readers can fully understand experimental design and evaluate the validity of the results. As written, it would not be possible to replicate this work. A sample size of 6 snakes, with 3 males and 3 females is not particularly robust, paticularly if there are sex differences, and may limit inferences drawn from the data.

Validity of the findings

This manuscript appears to give the erroneous impression that it is the first to explore habitat and thermal ecology in T. eques, which is incorrect. The authors do not use existing literature and MS theses on habitat use and ecology of T. eques from the southwestern U.S., and instead rely heavily on non-gartersnake habitat use studies as a proxy.

Additional comments

Overall, I thought this was a potentially very interesting study, with implications for conservation of a species that is federally listed as Threatened in the U.S. However, this study’s conceptual basis is lacking adequate review of existing US and Mexican T. eques studies. Critical details are also missing from the methods. Taken together, these issues made it challenging for me to critically evaluate the results and discussion.

---

## Round 0.2 · Major Revisions

Although the reviewer finds the ms. much improved, they still have many suggestions which will improve it even more. Therefore, please make the changes they have indicated in this round of reviews. Include a document outlining the changes you have made.

Reviewer 3 ·

Basic reporting

I have reviewed the work titled “Thermal ecology of Mexican garter snake (Thamnophis eques): temporal and spatial variations” submitted to the journal PeerJ. The manuscript has been substantially improved in many aspects from the last version reviewed. Therefore, I consider that is close to a more publishable version, once a few other aspects are fixed. Considering this, I have made specific suggestions arranged by line numbers throughout the manuscript, hoping that the authors consider my recommendations. Thank you and congratulations for this contribution to the thermal ecology of Mexican snakes.

Experimental design

No comment

Validity of the findings

Not comment

Additional comments

L1: I recommend adding the article “the” before “Mexican”
L30: Please consider using a synonym for one “show” in “our results show that T. eques showed…”
L39: Please abbreviate the generic epithet of “Thamnophis”
L44: Please consider changing “In, snake…” for “For instance, snake…” or “For example, snake…” I think they are better connectors.
L57-58: Given that this is the first time that T. eques is mentioned, the generic epithet should be spelled out.
L62: I think it should be “…air and water temperatures” instead of singular. This is because you mentioned two different types of environmental temperature. Additionally, it is possible that a dashed line is missing between “Macias Garcia”
L63-68: I suggest authors adding a documented example of snake species which Tb varies seasonally and spatially (i.e. between habitats/environments). By adding it, the authors will support their claim and give readers the idea of how and to what degree snake Tb changes regarding these factors.
L75-76: This example of Elaphe obsoleta is a similar one to I suggested previously, but I invite authors to develop it more. For example, does E. obsoleta select grasslands over forests because they are warmer environments? Alternatively, do female E. obsoleta prefer warmer microhabitats than males in both habitats? Again, this will give more background to readers regarding what kinds of aspects are evaluated.
L85-87: I suggest removing “with activity” to avoid redundancy with “bimodal annual activity”. Moreover, you are only mentioning one activity cycle (April to November) where I suspect are included the both activity cycles you are mentioning. Therefore, please specify both activity cycles if you are mentioning as “bimodal annual activity.” On the other hand, you are including the protection category of this Thamnophis species, which I consider is great to show how protected this snake species is in both countries. However, I think that something important is missing. For instance, what is the link between the protection category and thermal ecology in this particular snake species? This is especially true in southern and urban populations, which live in a region that experiences a continuous land-use change rate.
L88: I think “habitats” or “environments” will be better words than “resources”. Therefore, please consider this.
L91: I suggest adding an adjective to “biological model” probably “appropriate” or “suitable” would work well.
L98: I suggest defining when late fall and early winter occur, especially for readers outside Mexico.
L120-122: Through the manuscript, authors continuously use two different categories of tree canopy cover: when willow trees do not lose their leaves and when willow trees lose their leaves. I think it is a difficult and wordy categorization, and I offer an option to the authors to improve its usage. In this first mention, authors could established these categories with something like “…when tree canopy cover is continuous (i.e. when willow trees do not lose their leaves, May-September)… when tree canopy cover is absent (i.e. when willow trees lose their leaves, October-November)…”. Throughout the manuscript, the authors could simply mention, “When tree canopy cover was continuous” or “when tree canopy cover was absent,” instead of concentrating on whether trees lost their leaves or not. The important and suggested adjectives here are “continuous” and “absent” according to the way canopy cover is described, but I invited authors to use other adjectives if they considered more appropriate. Please consider this suggestion to improve categorization throughout the manuscript.
L136: Please remove the word before “environment” and change the first letter of “Mean” to lower case.
L137-138: Please correct the test name. It must be “Kolmogorov-Smirnov”. It does not make sense to me why you used this test, because as it is written, I understand that you compare the mean air temperature between habitats within each season. In such cases, the Mann-Whitney U test would perform better. Is just a doubt, and if it will be great, you could clarify it.
L152: I suggest changing the precision of the thermometer at the end of this paragraph and mentioning it only once, given that both Ta and Ts were measured using the same instrument.
L173: Please change the first letter of “Mean” to lower case.
L174-176: How exactly do you sexed snakes? I strongly suggest mentioning this given that there are different characters that one can use to do so. Additionally, which reference do you use to define adulthood? Adult males (SVL: 503.5 mm) are slightly smaller than adult females (SVL: 544.2 mm), so it is possible that SVL at sexual maturity differs between sexes; therefore, adulthood should be defined differently for each sex. Please explicitly define these aspects.
L202: Considering you refer to sample sizes here, I suggest changing “n values” for “sample sizes” for simplicity.
L207: I suggest adding analysis parameters to support your claims about no sexual differences as well as no significant effect of snake size on thermal traits. Probably, the p-values would be sufficient. Additionally, considering that ‘thermoregulation’ is the process of body temperature regulation, I think you probably mean body temperature. Therefore, please change “thermoregulation” to “body temperature” or “Tb”. However, if you mean all the temperatures you measured, I suggest using “thermal traits”.
L216: Which temperature remains constant: Ta or Ts?
L233: Here, the authors show six values, but in line 232 there are only five time intervals, so please make the necessary corrections to have the same number of time intervals and differences in Tb.
L241: As suggested in the previous paragraphs, please add analysis parameters to support your claims about similarities between habitats in Tb. Probably, the p-values would be sufficient.
L242-243: For simplicity, “spring, summer and autumn months” could be replaced by “all months and seasons”. So please consider it.
L273-274: What kind of cover do you mean: rocky cover or vegetative cover? Please specify it. Moreover, change “studio” for “study” and rewrite “…the T. eques Tb…”. In the latter case, probably “the Tb of T. eques’ may fit better.
L289-305: Throughout this paragraph, authors continuously mention “thermoregulatory strategy ‘or “thermoregulatory strategies’ and in line 296, they showed two examples of these strategies: “…time spent basking or under rocks…” However, considering the basic literature on thermoregulation in amphibians and reptiles (Huey and Slatkin, 1976; Adolph and Porter, 1993; Vitt and Caldwell, 2009), these are simply known as “behavioral mechanisms”. On the other hand, “thermoregulatory strategies” are defined considering if body temperature is similar to (passive thermoregulator) or different (active thermoregulator) from environmental temperatures. Considering this knowledge, I invite authors to change “thermoregulatory strategy”, “thermoregulatory strategies”, or “behavioral strategies” for “behavioral mechanisms”.
Huey RB, Slatkin M. 1976. Cost and benefits of lizard thermoregulation. The Quarterly Review of Biology 51: 363-384.
Adolph, S. C., and W. P. Porter. 1993. Temperature, activity, and lizard life histories. The American Naturalist 142:273–295.
Vitt LJ, Caldwell JP. 2009. Herpetology: An introductory biology of amphibians and reptiles. Third edition. Academic Press, San Diego, California.
L313: Please change the first letter of “Snake” to the lower case.
L318: Please change “occur in” for “from” and add, “occur” before “in the middle of…”
L320-331: I think the work of Grigg and Buckley (2013) would help to support this effect of phylogeny and space on thermal traits. So, please consider reviewing and adding it.
Grigg, J. W., & Buckley, L. B. (2013). Conservatism of lizard thermal tolerances and body temperatures across evolutionary history and geography. Biology Letters, 9: 20121056.
L355: Please change “prey terrestrial and aquatic” for “terrestrial and aquatic preys”
L369-370: Please change “…survival of gartersnake species…” for “…survival of this Gartersnake species”
L377: I recommend that authors avoid using subspecies, considering that this is an ecological study.
L380: Please remove “the” before the snake species name.
Table 1: The most-right column of this table have “n” column with empty spaces, I think that sample sizes are lacking. Please add the missing information.
Table 2: Please change “moths” for “months”. I suggest removing the first row and starting the table legend with “Pearson correlation coefficients…”. Additionally, given that the p-values of all correlations were equal, I recommend removing these values of the parentheses and adding a statement in the table legend with something like “P-values ≤ 0.0001 in all tests.”
Fig 2: The authors show here seven graphs, but I do not find any mention of each of these in any part of the manuscript. In the last version of the manuscript, I recommend using different colors for each line, but the authors claim that the lines are distinguishable. To the best of our knowledge, PeerJ does not charge color figures, but encourages authors to have some considerations when creating color figures. Therefore, I suggest once again that the authors should use colors for each line in the graphs or increase the size of the points for each line. The size of the points in the legend is visible, but the size of the points in the graphs is not visible.
Fig 3: I have the same suggestions of colors or increase the size points of the graphs, as in fig 2.

---

## Round 0.3 · accepted · Accept

Thank you for your attention to detail, and congratulations!